

# Feature screening for survival trait with application to TCGA high-dimensional genomic data

Jie-Huei Wang, Cai-Rong Li and Po-Lin Hou

Department of Statistics, Feng Chia University, Taichung, Taiwan

Corresponding author
Jie-Huei Wang,
jhwang@mail.fcu.edu.tw

## ABSTRACT

**Background:** In high-dimensional survival genomic data, identifying cancer-related genes is a challenging and important subject in the field of bioinformatics. In recent years, many feature screening approaches for survival outcomes with high-dimensional survival genomic data have been developed; however, few studies have systematically compared these methods. The primary purpose of this article is to conduct a series of simulation studies for systematic comparison; the second purpose of this article is to use these feature screening methods to further establish a more accurate prediction model for patient survival based on the survival genomic datasets of The Cancer Genome Atlas (TCGA).

**Results:** Simulation studies prove that network-adjusted feature screening measurement performs well and outperforms existing popular univariate independent feature screening methods. In the application of real data, we show that the proposed network-adjusted feature screening approach leads to more accurate survival prediction than alternative methods that do not account for gene-gene dependency information. We also use TCGA clinical survival genetic data to identify biomarkers associated with clinical survival outcomes in patients with various cancers including esophageal, pancreatic, head and neck squamous cell, lung, and breast invasive carcinomas.

**Conclusions:** These applications reveal advantages of the new proposed network-adjusted feature selection method over alternative methods that do not consider gene-gene dependency information. We also identify cancer-related genes that are almost detected in the literature. As a result, the network-based screening method is reliable and credible.

## INTRODUCTION

In high-dimensional genomic data, identifying important genes related to clinical survival traits is a challenging and important problem in the field of bioinformatics. The discovery of important biomarkers that explain the phenotype of interest is essential for the development of phenotype prediction models. In particular, contaminated data and right-censored survival outcomes make relevant feature screening more challenging.

In the era of high-throughput biology, the number of potential features/biomarkers could be much larger than the research sample size. In this case, it is well-known that use of preliminary feature screening can substantially improve the model selection performed by the regularization approach (*Fan & Lv, 2008*). Univariate feature screening for right-censored survival outcomes has been a challenging topic receiving much attention in the literature. *Edelmann et al. (2020)* recently provided a comprehensive review and some useful suggestions for univariate feature independent screening methods, and pointed out these screening methods for survival traits can be roughly divided into the following categories: (semi-)parametric-based approaches, non-parametric ranking-based approaches, model-free approached based on conditional survival functions, and distance correlation-based approaches. All univariate independent feature screening methods are based on certain statistics with specific model assumptions, which are calculated for each variable without considering other variables. This statistic measure is the so-called marginal utility. The feature screening procedure can then be performed by selecting important features based on their corresponding marginal utility. Since the marginal model is low-dimensional, the main advantage of the marginal model is its computational stability and conceptual simplicity. Therefore, marginal programs are still popular in the fields of bioinformatics.

However, outlier-contaminated biomarker data pose a further challenge to the survival prediction problem based on high-dimensional genetic/genomic data. As is known, the developmental process of disease is complicated and may involve the interaction of multiple genes; that is, epistasis. In other words, to effectively identify disease-related genes, it is necessary to make full use of biological network information. *Wu, Zhu & Feng (2018)* pointed out that ignoring gene-gene dependency information might lead to bias in gene screening. To this end, *Wang & Chen (2021)* developed a network-adjusted Kendall's tau measure for feature screening by incorporating gene-gene dependency network information and compared a network-adjusted measure to a partial-likelihood screening method (*Fan, Feng & Wu, 2010*; *Zhao & Li, 2012*) and inverse probability-of-censoring weighted (IPCW) Kendall's tau statistics (*Song et al., 2014*; *Wang & Chen, 2020*). They proved that the network-adjusted Kendall's tau measurement method is superior to these two methods in variable screening for most of the network structures considered in terms of the average number of true predictors contained in the selected model and the minimum of model size.

In this article, we intend to perform a systematic comparison for these advanced feature screening methods and apply them to The Cancer Genome Atlas (TCGA, *The Cancer Genome Atlas Research Network, 2008*) survival genomic data to develop a more accurate prediction model for patient survival. The simulation studies under various scenarios are conducted to compare the performance of new network-adjusted IPCW Kendall's tau measure with several commonly used univariate independent feature screening methods. In the application of real data, we demonstrate that the new network-adjusted feature screening approach leads to more accurate survival prediction than alternative methods that do not account for feature network information or outlier-contamination. We also determine biomarkers that are associated with clinical survival outcomes of patients with

esophageal carcinoma (ESCA), pancreatic adenocarcinoma (PAAD), head and neck squamous cell carcinoma (HNSCC), lung adenocarcinoma (LUAD), and breast invasive carcinoma (BRCA) using TCGA genetic data.

## MATERIALS AND METHODS

### Data structure and methods partial review

We consider a study with n independent subjects. For a subject i, suppose that there are p genes expression $(x_{i1}, \ldots, x_{ip})'$ related to clinical survival outcomes $T_i$. Note that the number of the genes is far greater than the sample size n. Usually, the survival outcome is subject to censoring, so we define $C_i$ as censoring time, and use $\delta_i$ as the indicator of whether the survival time of subject i is censored, then define $V_i = \min(T_i, C_i)$ as observed survival time.

We list the common and effective survival feature screening methods, and provide readers with an overview summarized in Table 1. *Edelmann et al. (2020)* provided an R package "*MVS*" that can be downloaded from https://github.com/thomashielscher/MVS, which contains the first six screening methods discussed in Table 1, and R codes for the last two screening methods discussed in Table 1 are available at https://figshare.com/articles/software/CODE/16677070, laying the foundation for meaningful comparison.

### TCGA cancer data source

TCGA RNA-Seq expression data and phenotypic data including survival time and censoring status data can be downloaded from the R package 'TCGAbiolinks' (*Colaprico et al., 2016*), or 'UCSCXenaTools' (*Wang & Liu, 2019*). The TCGA ESCA, PAAD, LUAD, and BRCA genomic data with survival traits analyzed during this study are all available at Figshare: https://figshare.com/articles/dataset/DATA/16677619. The TCGA HNSCC genomic data can be downloaded from the R package "GEInter" (*Wu, Qin & Ma, 2021*).

### Evaluation performance in the simulation study

In performance measurement, we report the percentiles of the minimum model size (MMS) statistics among 200 replications through violin plot to view the distribution of MMS data and its summary statistics, where MMS is the minimum size of a selected model, including underlying effective predictors. MMS measures the complexity of the selected model and reflects the accuracy of the screening process; a smaller MMS value indicates the higher accuracy of feature screening. We note that a violin chart can be constructed through the "*vioplot*" R package (*Adler & Kelly, 2021*). We also performed additional simulation studies to investigate the survival time prediction errors by giving the average number of $c$-index (*Harrell, Lee & Mark, 1996*) among 200 replications as a function of the number of selected features for each method. The $c$-index metric compares the subjects' predicted survival time rankings with their real survival time rankings. A larger $c$-index indicates better prediction accuracy.

For our comparative study, we assess the similarity of different screening methods in terms of the list of detected features. To this end, for each simulation run, the Jaccard

**Table 1 Reviews on univariate feature screening for survival outcomes (a partial list).**

| Citation | Class/Method | Description |
|---|---|---|
| *Fan, Feng & Wu (2010)* and *Zhao & Li (2012)* | (semi-)parametric-based approach/partial likelihood (PL) | The PL approach takes the maximum value of the corresponding marginal Cox's partial likelihood as marginal utilities to rank the predictors. |
| *Saldana & Feng (2018)* | (semi-)parametric-based approach/sure independence screening (SIS) | The SIS approach is an *ad hoc* approach, which takes Pearson correlations between predictors and survival times as marginal utilities to rank the predictors. |
| *Gorst-Rasmussen & Scheike (2012)* | (semi-)parametric-based approach/feature aberration at survival times (FAST) | The FAST approach proposes a semi-parametric independent screening method for survival data which are described by single-index hazard rate models. |
| *Chen, Chen & Wang (2018)* | distance correlation-based approach/robust censored distance correlation screening (RCDCS) | The RCDCS approach takes the robust distance correlation (*Zhong et al., 2016*) by replacing the survival outcomes and predictors by their corresponding cumulative distribution functions' Kaplan–Meier estimator and empirical distribution function as marginal utilities to rank the predictors. |
| *Chen, Chen & Wang (2018)* | distance correlation-based approach/composite robust censored distance correlation screening (CRCDCS) | The CRCDCS approach modifies the robust distance correlation through the composite quantile distance correlation of *Chen, Chen & Liu (2019)* to the right-censored scenario by redistributing the masses of censored observations to the right with the indicator function being involved. |
| *Harrell, Lee & Mark (1996)* | non-parametric ranking-based approach/Harrell's concordance index (CINDEX) | The CINDEX approach takes the C-index as marginal utilities to rank the predictors. |
| *Song et al. (2014)* and *Wang & Chen (2020)* | non-parametric ranking-based approach/inverse probability-of-censoring weighted (IPCW) Kendall's tau | The IPCW Kendall's tau approach takes Kendall's tau rank correlation as marginal utilities to measure the association between survival trait and biomarkers, which uses the IPCW technique to accommodate right-censored survival outcomes. |
| *Wang & Chen (2021)* | network-based approach/: IPCW-tau (NPN-MB) | The NPN-MB approach modifies the IPCW Kendall's tau measure (*Wang & Chen, 2020*) to incorporate gene–gene dependency network information using the technique of Google's PageRank Markov matrix. The NPN-MB approach using the nonparanormal procedure (*Liu, Lafferty & Wasserman, 2009*) to transform the original predictors to follow multivariate normal distribution, and then using the MB method (*Meinshausen & Buhlmann, 2006*) to estimate the sparse precision matrix. |

index ($J(A, B) = |A \cap B|/|A \cup B|$) of the true feature list is calculated by selection of by two methods under a specific model size of 500. The average Jaccard index of 200 simulation repetitions is used as a measure of similarity between the two methods; in this way, the similarity matrix of all the screening methods under consideration can be constructed and visualized, for example, using the "*corrplot*" R package (*Wei & Simko, 2017*).

In addition, we calculate the overlap coefficient ($O(A, B) = |A \cap B|/\min(|A|, |B|)$) set similarity analysis of features selected by each method with a ground truth set of predictors. The average number of overlap coefficient index among 200 replications as a function of the number of selected features for each method is used as a measure of similarity between the feature screening method and the ground truth set of predictors. A larger overlap coefficient index indicates highly similarity with a ground truth set of predictors.

## Simulation scenarios

For the first simulated settings, we follow the simulation settings of *Song et al. (2014)*, and generate a cohort of 500 subjects. Each subject's survival time follows the linear transformation model

$$H(T_i) = -x_i'\beta_0 + \varepsilon,$$

where $H(t) = \log(0.5(e^{2t} - 1))$, the covariates x jointly follow a 2,000-dimensional multivariate standard normal distribution with the first-order autoregressive (AR(1)) structure that is $\mathrm{corr}(x_{.j}, x_{.k}) = 0.5^{|j-k|}$. The distribution of the error term $\varepsilon$ follows a standard extreme value distribution, which corresponds to a proportional hazards (PH) model.

The true regression coefficient vector is sparse:

$$\beta_0 = (-1, \; -0.9, \; 0.5, \; 0.8, \; 0.6, 0_5, 0.3,$$
$$0.7, -0.8, -0.5, -1, \; 0_5, -2, \; 1, \; 0, \; -0.5, -2, 0_{1,975}),$$

so the underlying survival model has 14 true predictors. In the first simulated settings, only linear relationships were assumed with true parameter vector. The censoring time distribution follows a uniform distribution $U(a, b)$, where $(a, \; b)$ is chosen to control the censoring rate at about 30% (light censoring), 50% (middle censoring) and 70% (heavy censoring) respectively. Moreover, we consider the setting where, with a probability of 0.1, the covariates may be contaminated by outliers produced by a *t* distribution with two degrees of freedom.

For the second simulated settings, we follow the simulation settings of *Edelmann et al. (2020)*. The simulated settings are the same as the first simulated settings except for the relationships of true parameter vector, *i.e.*,

$$H(T_i) = -z_i'\beta_0 + \varepsilon,$$

where $z_i' = (g_1(x_{i1}), \ldots, g_p(x_{ip}))$, we assume $g_1(x) = \beta_1|x|$, $g_2(x) = \beta_2|x|, g_4(x) = \beta_4 x^2$, $g_5(x) = \beta_5 1(x > 0)$, and all other j, we assume $g_j(x) = \beta_j x$.

For the third simulated settings, we follow the simulation settings of *Wang & Chen (2021)*. The simulated settings are the same as the first simulated settings except for the correlation structure of the variables and the true regression coefficient. We generate the covariates jointly following a 2,000-dimensional multivariate standard normal distribution with different network structures, including "hub", "band", "cluster", and "scale-free". These network structures can be generated by the "*huge*" package with *huge.generator* function (*Zhao et al., 2012*) and the true regression coefficient vector is defined as

$$\beta_0 = (-1.5, \; -1.5, \; 1.5, \; 1.5, \; 1.5, 0_5, 1.5,$$
$$1.5, -1.5, -1.5, -1.5, \; 0_5, -1.5, \; 1.5, \; 1.5, \; -1.5, -1.5, 0_{1,975})$$

For each simulation scenario, we perform 200 replications to investigate the numerical performances of different methods.

### Survival prediction measure in real data application

To evaluate the performance of survival prediction, we report three measures of prediction accuracy: the *c*-index, deviance, and the number of selected features (NOSF) metrics. Larger *c*-index/smaller deviance and number of selected features indicates better prediction accuracy. Since the *c*-index metric can be used to compare a subject's predicted survival time ranking with the true survival time ranking. And, the deviance metric is defined as

$$D = -2\left(logL\left(\hat{\boldsymbol{\beta}}\right) - logL(0)\right),$$

where $logL\left(\hat{\boldsymbol{\beta}}\right)$ is the log partial likelihood function of Cox's model from the testing set of the data, $\hat{\boldsymbol{\beta}}$ is an estimator of the penalized Cox's regression with the MCP penalty parameter in a prediction model obtained from the training set of the data, and $logL(0)$ is the log partial likelihood function of Cox's null model from the testing set of the data, where all predictors are assumed not related to the true survival time.

The deviance metric can therefore be considered as a comparison between the survival prediction model and null model (no predictors considered). The deviance metric is also a suitable survival prediction criterion.

Finally, we choose the number of selected features metric as a precision criterion; the reason is that the feature screening approach is used to find parsimonious precision models. Meaning, if we are modeling two sets of features with the same predictive accuracy, we want to choose the model that uses the features, as a smaller number of selected features are easier to interpret/evaluate in follow-up studies about biological function.

## RESULTS

### Simulation studies

In simulation studies, a series of simulation studies are conducted to investigate the performance of the existing feature screening methods for survival trait in identifying true associated predictors and survival prediction errors.

The simulation results are summarized in Figs. 1–6 and Figs. S1–S12. Note that Figs. 1–3 indicate the performances for the simulation study 1 with AR(1) structure; Figs. 4–6 indicate the performances for the simulation study 2 with nonlinear structure; Figs. S1–S3 indicate the performances for the simulation study 3 with band structure; Figs. S4–S6 indicate the performances for the simulation study 3 with hub structure; Figs. S7–S9 indicate the performances for the simulation study 3 with cluster structure; and Figs. S10–S12 indicate the performances for the simulation study 3 with scale-free structure. We note that the "IPCW-tau (MB)" method is also an IPCW Kendall tau method of network adjustment, but it uses the original predictors without using nonparanormal procedure to transform them.

From simulation studies 1, 2, and 3 with band structure, we see that the IPCW-tau (NPN-MB) method outperforms all alternative methods in terms of the overlap coefficient (top three panels of Figs. 1, 4, S1), and MMS measure (Figs. 2, 5, S2). Overall, the variable

Peer

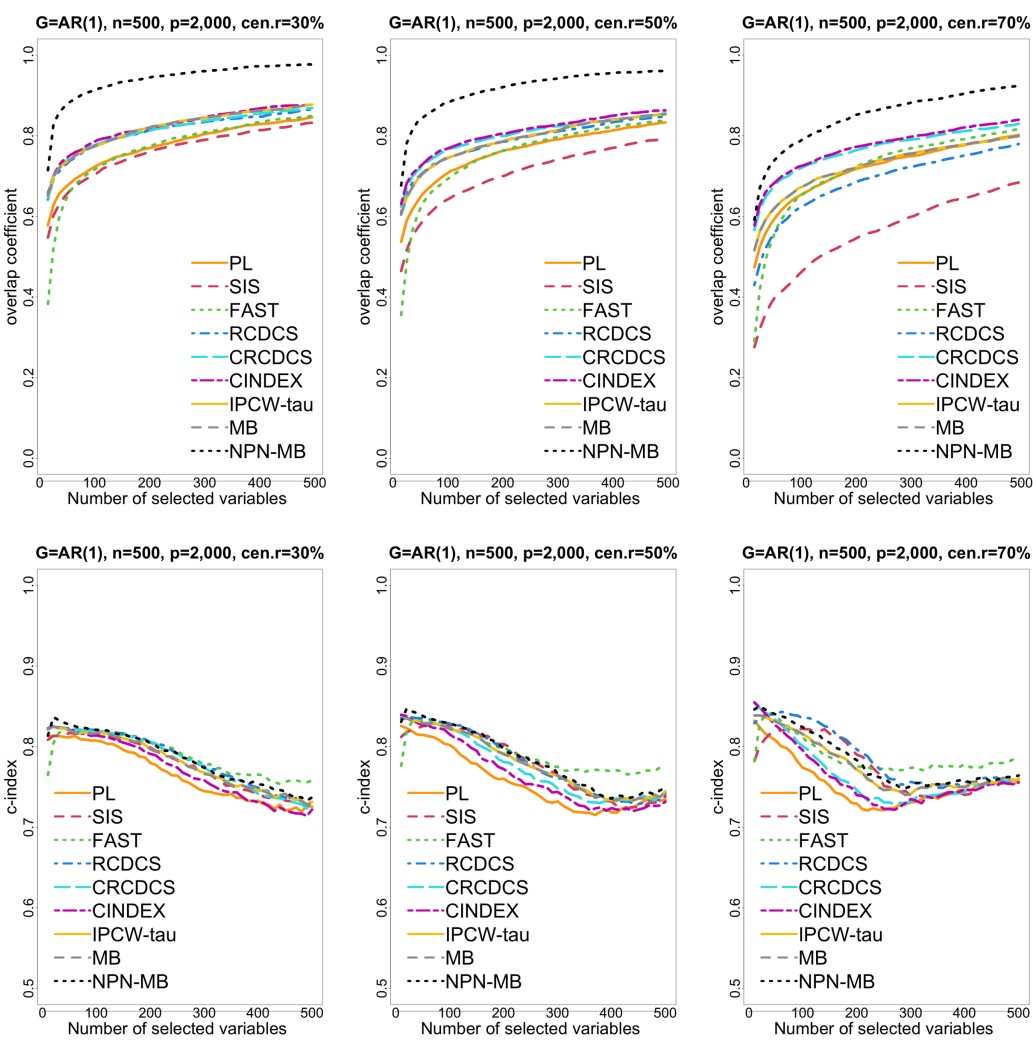

**Figure 1** **The multi-panel figure contains the mean number of overlap coefficient (top three panels) and *c*-index (bottom three panels) among 200 replications by the number of selected features for the simulation 1 with AR(1) structure based on PH model.** The left, medium, and right plots are based on censoring rates of 30%, 50%, and 70%, respectively. A larger mean number of overlap coefficient indicates highly similarity with a ground truth set of predictors, and a larger c-index indicates better prediction accuracy. Note that the underlying survival model has 14 true predictors.

selection accuracy of the IPCW-tau (NPN-MB) method is substantially better than all other methods. From bottom three panels of Figures 1, 4, S1, the FAST method has a higher *c*-index when the number of selected variables is larger, and the IPCW-tau (NPN-MB) method outperforms most alternative methods when the number of selected variables is medium or small.

In simulation study 3 with hub structure, we see that the IPCW-tau (NPN-MB) method performs the best when the number of selected variables is larger or medium in terms of the overlap coefficient (top three panels of Fig. S4), then the CINDEX method performs the best when the number of selected variables is small. The *c*-index measure patterns (bottom three panels of Fig. S4) are similar to these in the previous simulation studies.

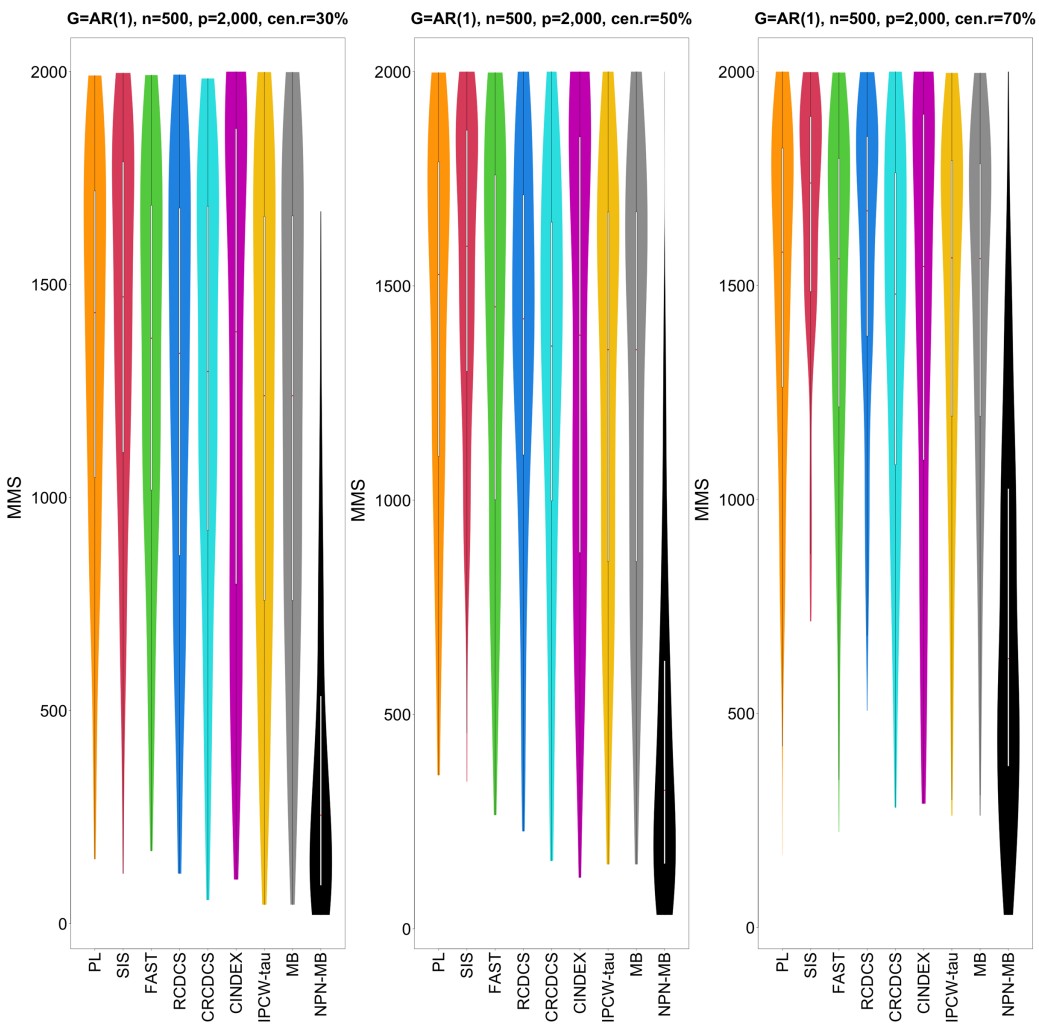

**Figure 2 The violin chart of minimum of model size (MMS) measure among 200 replications for the simulation study 1 with AR(1) structure based on PH model.** The left, medium, and right plots are based on censoring rates of 30%, 50%, and 70%, respectively. A smaller MMS value indicates the higher accuracy of feature screening.

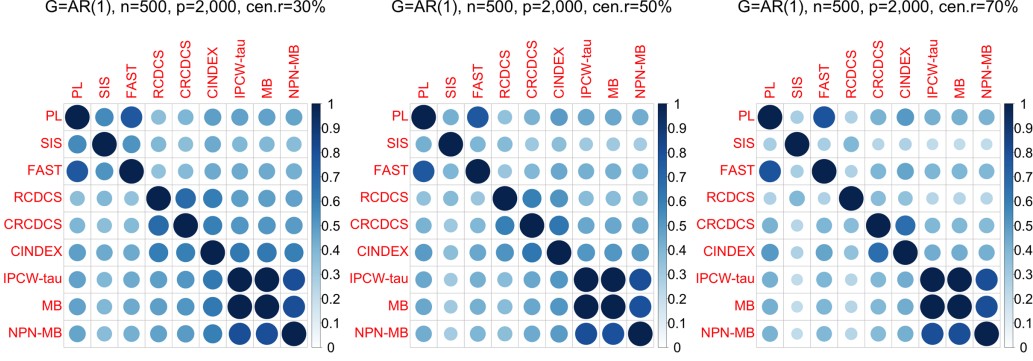

**Figure 3 The average of Jaccard index among 200 replications for the simulation study 1 with AR(1) structure based on PH model.** The left, medium, and right plots are based on censoring rates of 30%, 50%, and 70%, respectively.

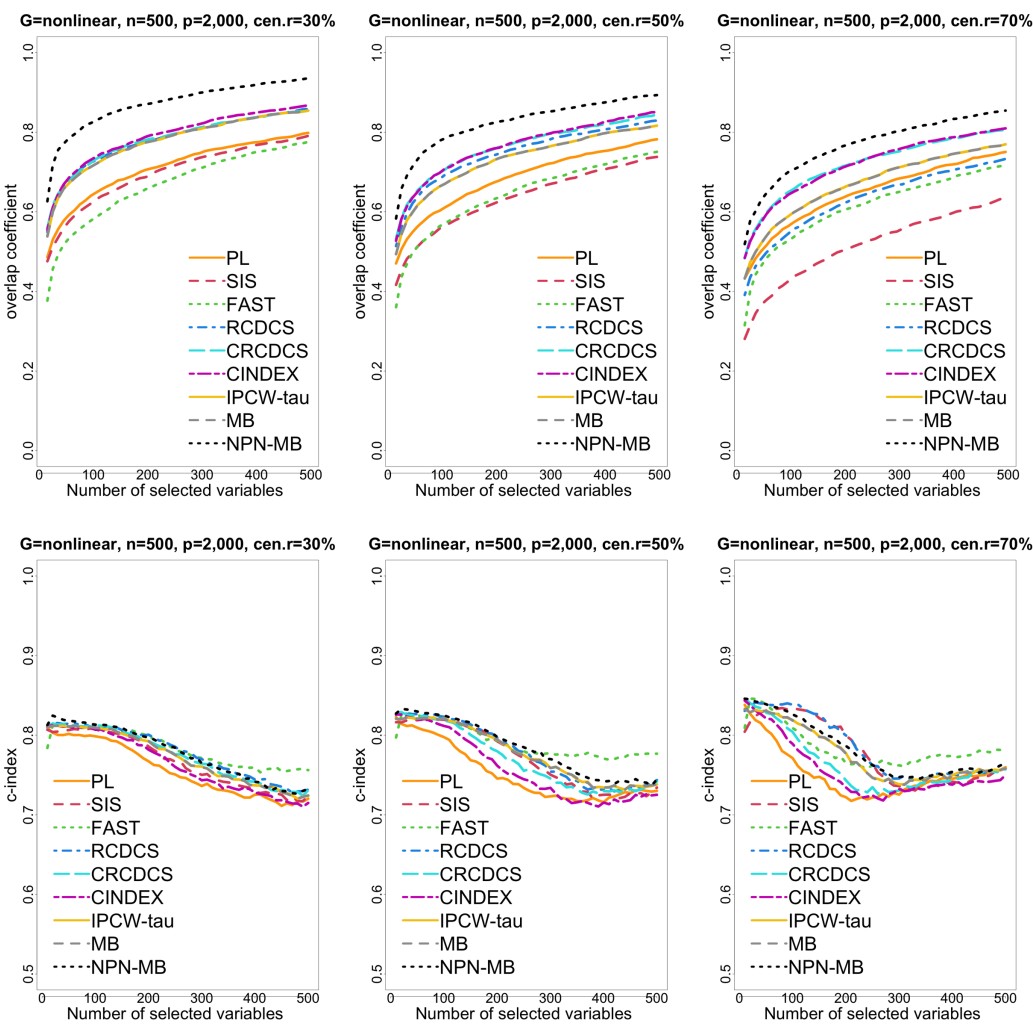

**Figure 4** **The multi-panel figure contains the mean number of overlap coefficient (top three panels) and *c*-index (bottom three panels) among 200 replications by the number of selected features for the simulation two with nonlinear structure based on PH model.** The left, medium, and right plots are based on censoring rates of 30%, 50%, and 70%, respectively. A larger mean number of overlap coefficient indicates highly similarity with a ground truth set of predictors, and a larger c-index indicates better prediction accuracy. Note that the underlying survival model has 14 true predictors.

On the MMS measure metric (Fig. S5), the IPCW-tau (NPN-MB) method performs the best.

In simulation study 3 with cluster and scale-free structures, we see that the CINDEX method performs the best when the censoring rate is high in terms of the overlap coefficient (top three panels of Figs. S7, S10), then the IPCW-tau (NPN-MB) method performs the best when the censoring rate is medium or small. The *c*-index measure patterns (bottom three panels of Figs. S7, S10) are similar to these in the previous simulation studies. In the MMS measure metric (Figs. S8, S11), the performance of IPCW-tau (NPN-MB) method is better than other methods.

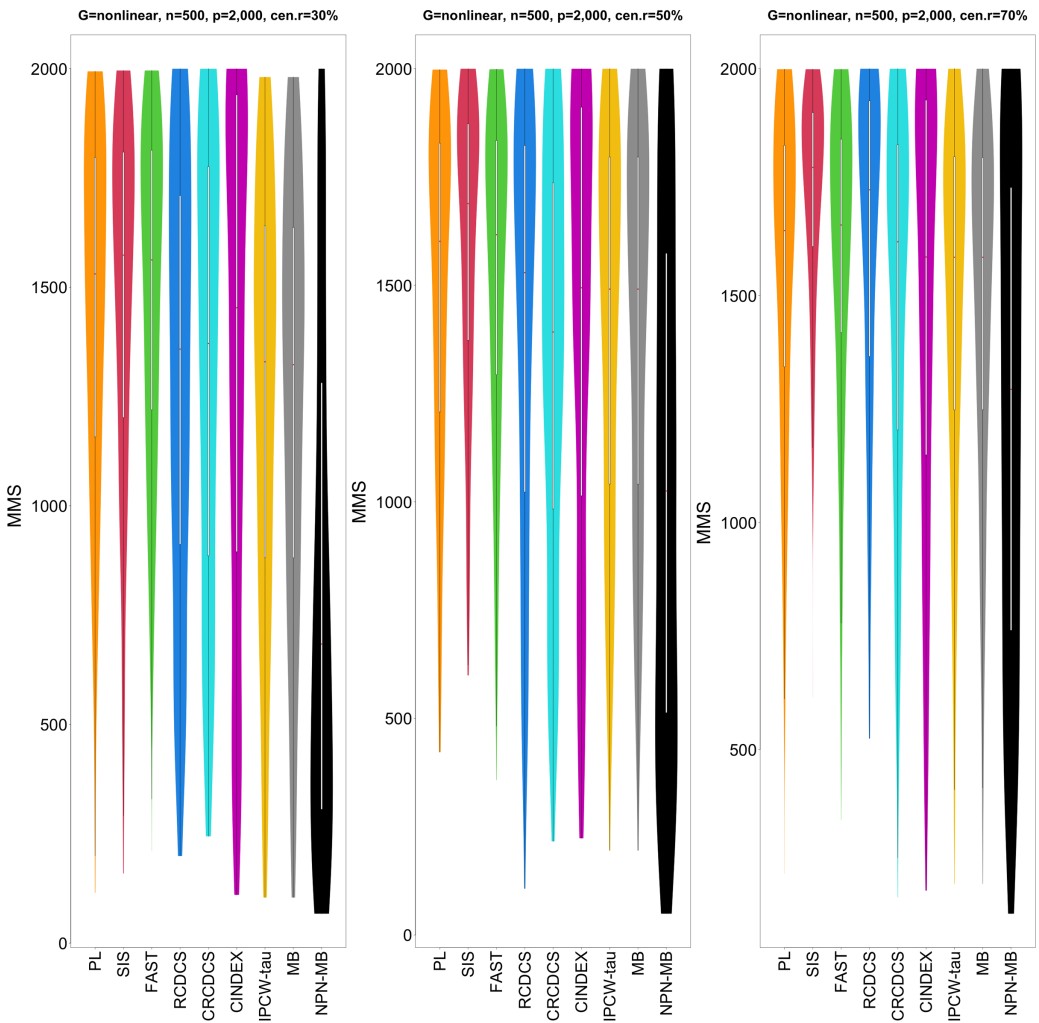

**Figure 5 The violin chart of minimum of model size (MMS) measure among 200 replications for the simulation study 2 with nonlinear structure based on PH model.** The left, medium, and right plots are based on censoring rates of 30%, 50%, and 70%, respectively. A smaller MMS value indicates the higher accuracy of feature screening.

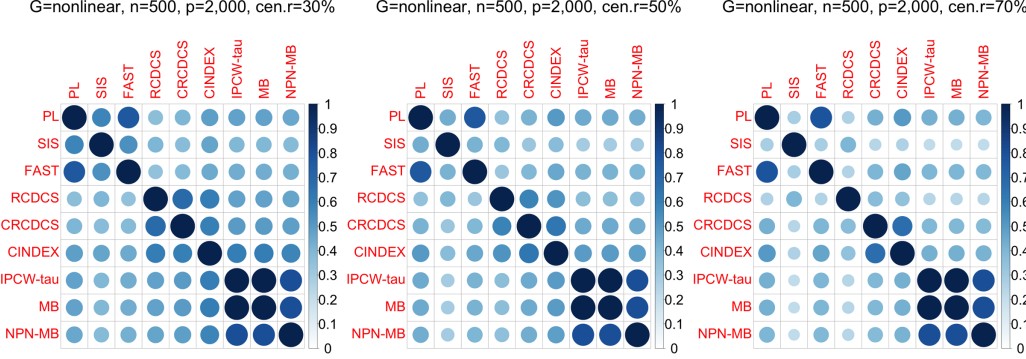

**Figure 6 The average of Jaccard index among 200 replications for the simulation study 2 with nonlinear structure based on PH model.** The left, medium, and right plots are based on censoring rates of 30%, 50%, and 70%, respectively.

**Table 2 Results (median of prediction accuracy of different methods in the TCGA ESCA data over five random splits of 294:74 training/test sets).**

|          | PL      | SIS     | FAST    | RCDCS    | CRCDCS  | CINDEX | IPCW-tau | NPN-MB   | Ordinary-MCP | *Du et al. (2021)* |
|----------|---------|---------|---------|----------|---------|--------|----------|----------|--------------|--------------------|
| Deviance | 17.4067 | −5.6785 | −6.2444 | −2.79612 | −3.7387 | 2.7191 | 1.0834   | −19.7218 | 474.4513     | 0.3548             |
| *c*-index | 0.6542 | 0.6314  | 0.6690  | 0.6866   | 0.6943  | 0.7324 | 0.6236   | 0.7380   | 0.8466       | 0.5450             |
| NOSF     | 14      | 4       | 13      | 8        | 8       | 12     | 12       | 9        | 36           | 2                  |

**Note:**

All feature screening methods and a published biomarker genes model are applied together with the MCP penalized Cox regression.

According to the average of Jaccard similarity index (Figs. 3, 6, S3, S6, S9, S12), we find that screening methods belonging to the same category have higher similarity than screening methods not belonging to the same category.

A further simulation study is conducted under the scenario where the survival time follows a proportional odds (PO) model (*i.e.*, the error term follows a standard logistic distribution) and all other settings are the same as those in the previous simulation study. We still obtain similar numeric results and the proposed IPCW-tau (NPN-MB) method has a better performance than the alternative methods at most gene structures. These correspondence figures are all available at Figshare: https://figshare.com/articles/figure/Survival_Feature_Screening_based_on_proportional_odds_model/17089013.

## Real data application with TCGA ESCA data

After excluding patients with missing survival time data, our analysis is focused on the subset of the TCGA ESCA data with 368 patients and 20,501 gene expression variables. The censoring rate in the data is about 58%. As the number of disease-associated biomarkers is not expected to be large, the top 2,000 genes with the smallest $p$-values based on marginal Cox's model are selected for downstream analysis. We take five random splits of the whole data into 294:74 training/test sets of the data to evaluate the performance of all methods for survival prediction in the TCGA ESCA data.

According to the procedure of *Wang & Chen (2021)*, we apply eight screening methods, "PL", "SIS", "FAST", "RCDCS", "CRCDCS", "CINDEX" "IPCW-tau", "IPCW-tau (NPN-MB)", to the TCGA ESCA data. After grid search from the top 10 to the top 300 ranked genes, the best overall prediction performance of all methods is attained by using the top 150 genes, so the top-ranked 150 predictors are selected as the candidate covariates for each method and the Cox's regression model with the candidate covariates and the MCP penalty (*Zhang, 2010*) is applied to the training data to establish the final prediction model. Besides, the MCP-penalized Cox model with the top 2,000 genes selected by the univariate Cox's test is applied to the training data to build the prediction model. We also take the published biomarker genes (*DNAJB1, BNIP1, VAMP7, TBK1*) related to ESCA (*Du et al., 2021*) as a survival prediction model to make comparisons.

The prediction accuracy performances for different methods are evaluated and the numerical results are provided in Table 2 that reports the median of the survival prediction results among five treatments. Overall, we can see that the proposed IPCW-tau (NPN-MB)

**Table 3 Selected genes with their correspondence estimate by IPCW-tau (NPN-NB) screening procedure with MCP penalty for the whole TCGA ESCA data.**

| gene | Estimate | Citation |
|---|---|---|
| ATRX | 0.7686 | |
| C16orf80 | 0.9168 | |
| C16orf87 | −0.2953 | |
| FAM189A2 | −0.1655 | |
| GFPT1 | 0.9318 | *Zhang et al. (2020b)* |
| HNRNPC | 1.1911 | *Xu, Pan & Pan (2020)* |
| MAF | 0.4708 | |
| NEK1 | −0.9591 | |
| OSTM1 | 0.7367 | |
| TSKU | −0.3521 | |

method outperforms the alternative methods for survival prediction in the TCGA ESCA test data.

In addition, we apply the proposed IPCW-tau (NPN-MB) method for whole data to identify several important biomarker genes and estimate the correspondence parameters by penalized Cox's regression model with the MCP penalty. Please see Table 3 for the list of selected associated predictors with their correspondence weights. We identify ten genes and find the two genes *(GFPT1, HNRNPC)* genes that are related to ESCA in the literature (*Zhang et al., 2020b*; and *Xu, Pan & Pan, 2020*).

## Real data application with TCGA PAAD data

After excluding patients with missing survival time data, our analysis is focused on the subset of the TCGA PAAD data with 178 patients and 20,501 gene expression variables. The censoring rate in the data is about 48%. As the number of disease-associated biomarkers is not expected to be large, the top 2,000 genes with the smallest $p$-values based on marginal Cox's model are selected for downstream analysis. We take five random splits of the whole data into 142:36 training/test sets of the data to evaluate the performance of all methods for survival prediction in the TCGA PAAD data.

According to the procedure of *Wang & Chen (2021)*, we apply eight screening methods, "PL", "SIS", "FAST", "RCDCS", "CRCDCS", "CINDEX" "IPCW-tau", "IPCW-tau (NPN-MB)", to the TCGA PAAD data. After grid search from the top 10 to the top 300 ranked genes, the best overall prediction performance of all methods is attained by using the top 20 genes, so the top-ranked 20 predictors are selected as the candidate covariates for each method, and the Cox's regression model with the candidate covariates and the MCP penalty (*Zhang, 2010*) is applied to the training data to establish the final prediction model. Besides, the MCP-penalized Cox model with the top 2,000 genes selected by the univariate Cox's test is applied to the training data to build the prediction model. We also take the published biomarker genes *(CDKN2A, TP53, TTN, KRAS)* related to PAAD (*Baek & Lee, 2020*) as a survival prediction model to make comparisons.

**Table 4 Results (median of prediction accuracy of different methods in the TCGA PAAD data over 5 random splits of 142:36 training/test sets).**

|  | PL | SIS | FAST | RCDCS | CRCDCS | CINDEX | IPCW-tau | NPN-MB | Ordinary-MCP | *Baek & Lee (2020)* |
|---|---|---|---|---|---|---|---|---|---|---|
| Deviance | −6.3363 | −2.3062 | −7.1140 | 2.5505 | −6.31712 | −5.1000 | −3.3673 | −9.7919 | 887.5797 | −4.4676 |
| c-index | 0.6834 | 0.6457 | 0.6608 | 0.5955 | 0.6774 | 0.6387 | 0.6538 | 0.6834 | 0.5290 | 0.7048 |
| NOSF | 5 | 4 | 3 | 6 | 6 | 6 | 5 | 2 | 39 | 1 |

**Note:**
All feature screening methods and a published biomarker genes model are applied together with the MCP penalized Cox regression.

**Table 5 Selected genes with their correspondence estimate by IPCW-tau (NPN-NB) screening procedure with MCP penalty for the whole TCGA PAAD data.**

| gene | Estimate | Citation |
|---|---|---|
| *MET* | 0.5718 | *Li et al. (2021)*, *Huang et al. (2021)*, *Wu et al. (2019a)*, *Vanderwerff et al. (2019)*, and *Li et al. (2019)* |
| *ZMAT1* | −0.1422 | |

The prediction accuracy performances for different methods are evaluated and the numerical results are provided in Table 4 that reports the median of the survival prediction results among five folds. Overall, we can see that the proposed IPCW-tau (NPN-MB) method outperforms the alternative methods for survival prediction in the TCGA PAAD test data.

In addition, we apply the proposed IPCW-tau (NPN-MB) method for whole data to identify several important biomarker genes and estimate the correspondence parameters by penalized Cox's regression model with the MCP penalty. Please see Table 5 for the list of selected associated predictors with their correspondence weights. We identify two genes (*MET*, *ZMAT1*) and find the *MET* gene that is related to PAAD in the literature (*Li et al., 2021*; *Huang et al., 2021*; *Wu et al., 2019a*; *Vanderwerff et al., 2019*; and *Li et al., 2019*).

Finally, the analysis results for TCGA HNSCC, TCGA LUAD, and TCGA BRCA data are provided in supplementary materials. Note that we take the published biomarker genes (*GIMAP6, SELL, TIFAB, KCNA3, CCR4*) related to HNSCC (*Ran et al., 2021*); (*ALK, BRAF, EGFR, ROS1*) related to LUAD (*Chen et al., 2021*); (*TMEM190, TUBA3D, LYVE1, LILBR5, CD209*) related to BRCA (*Liu et al., 2019*) as a survival prediction model to make comparisons. We identify nine genes and find the four genes (*PITPNM3, MXD4, ABCB1, BATF*) that are related to HNSCC in the literature (*Aravind et al., 2021*; *Wu et al., 2019b*; *da Silva et al., 2021*; *Duz & Karatas, 2021*; *Wang et al., 2020*; and *Wen et al., 2015*). We identify fifteen genes and find the seven genes (*EPB41L5, INPP5J, KRT16, MS4A1, MYLIP, PEBP1, SFTPB*) that are related to LUAD in the literature (*Li et al., 2020a*; *Zhang et al., 2020a*; *Yuanhua et al., 2019*; *Song et al., 2020*; *Liu et al., 2021b*; *Li et al., 2020b*; *Zhang et al., 2021*; *Cao et al., 2021*; and *Zhang et al., 2019*). We identify ten genes and find the four genes (*EDA2R, PCMT1, QPRT, SKP1*) that are related to BRCA in the literature (*Liu, Kain & Wang, 2012*; *Kyritsis et al., 2021*; *Liu et al., 2021a*; and *Tian*

*et al., 2020*). The proposed IPCW-tau (NPN-MB) method consistently performs well in these cancer datasets (refer to Tables S2, S4, and S6).

## CONCLUSIONS AND DISSCUSSIONS

The identification of cancer-related genes in high-dimensional genetic/genomic data is a challenging and important issue. In particular, right-censored survival outcomes and contaminated biomarker data make relevant feature screening difficult. A two-step statistical algorithm is used to achieve this (*Fan & Lv, 2008*). The first step is preliminary feature screening to identify biomarkers that may be associated with cancer, then the regularization approach is used to conduct the final variable selection and parameter estimation simultaneously.

The first purpose of this article is to conduct a systematic simulation study to validate the performance of the advanced feature screening methods in variable selection and survival prediction error. We prove that for most types of gene structures, the performance of the new network-adjusted feature screening method is better than most effective univariate independent feature screening methods. The second purpose of this article is to establish a survival prediction model for TCGA survival genomic data. We prove that, compared with alternative methods that do not consider feature network information or outlier-contamination, and the published biomarker genes models, the new network adjustment feature screening method can lead to more accurate survival prediction, and determine biomarkers that are associated with clinical survival outcomes of patients with ESCA, PAAD, HNSCC, LUAD and BRCA using TCGA genetic data. These applications show that the new network-adjusted feature selection method performs well and outperforms the existing popular univariate independent feature selection methods and the published biomarker genes models. We have also identified cancer-related genes almost detected in the literature. Accordingly, the new network-based screening method is reliable and credible. R codes for the simulation studies and real data are available at Figshare: https://figshare.com/articles/software/CODE/16677070.

According to simulation studies for the *c*-index measure, we observe that the FAST method has a higher *c*-index when the number of selected variables is larger, and the IPCW-tau (NPN-MB) method outperforms most alternative methods when the number of selected variables is medium or small. Although in the real data application, the *c*-index measure of the IPCW-tau (NPN-MB) method is not always the best among all considered feature screening methods or the published biomarker gene models, the IPCW-tau (NPN-MB) method still outperforms most alternative methods by the different prediction metrics. In addition, we observe every method can also identify the biomarker genes that are related to TCGA cancer in the literature. However, according to the simulation studies and real data analysis, we can still infer that the IPCW-tau (NPN-MB) method has better performance in both variable selection and survival prediction. To this end, the IPCW-tau (NPN-MB) method is a good choice for developing a survival prediction model.

The main purpose of *Wang & Chen (2021)* is to develop the advanced network-based Kendall's tau feature screening, and in simulation studies, only compared a network-based

measure to partial likelihood screening method and IPCW Kendall's tau statistics, not provide a systematic comparison for some popular advanced feature screening methods with survival outcome. As a consequence, the first purpose of this article was to review multiple screening approaches systematically and make comparisons under the various simulated scenarios with more evaluation performance like *c*-index for prediction errors, overlap coefficient index, and the Jaccard index. Moreover, in real data applications, *Wang & Chen (2021)* apply a few feature screening methods to only two real data. We apply more feature screening methods and the published gene signature models to five TCGA cancer genomic data, and provide an optimal survival prediction model for patients. Furthermore, we also provide the selected biomarker genes with their correspondence weights, which has meaning for clinical significance.

In the real data application, we adopt the hard thresholding rule proposed by *Fan & Lv (2008)* to select the candidate set of predictors; that is, after ranking the predictors using a certain correlation measure, we select a prefixed number of top-ranked predictors as our candidate model. Several alternative strategies for thresholding rule can be considered, such as the soft thresholding rule proposed by *Zhu et al. (2011)*, a method based on the control of the false-positive rate or false discovery rate by *Zhao & Li (2012)*, and a method based on multiple testing procedure by *Song et al. (2014)*. In addition, we assume that the number of cancer-related biomarkers will not be large, so we select the top 2,000 genes with the smallest *p*-value for downstream analysis based on the marginal Cox's model. Different candidate models lead to different survival prediction models. How to define the number of cancer-related biomarkers for downstream analysis is a key and interesting issue.

There are several public human databases, like METABRIC, TCGA, NCDB, and GEO. These useful databases can be utilized to determine the reproducibility of our findings. We consider that meta-analysis can be performed for discovery and validation of survival biomarker genes (*Xu et al., 2020*), which is worthy of further research and will be studied in our future work.

## LIST OF ABBREVIATIONS

| | |
|---|---|
| **TCGA** | The Cancer Genome Atlas |
| **ESCA** | esophageal carcinoma |
| **PAAD** | pancreatic adenocarcinoma |
| **HNSCC** | head and neck squamous cell carcinoma |
| **LUAD** | lung adenocarcinoma |
| **BRCA** | breast invasive carcinoma |
| **PL** | partial likelihood |
| **FAST** | feature aberration at survival times |
| **SIS** | sure independence screening |
| **RCDCS** | robust censored distance correlation screening |
| **CRCDCS** | composite robust censored distance correlation screening |
| **CINDEX** | Harrell's concordance index |

| | |
|---|---|
| **IPCW** | inverse probability-of-censoring weighted |
| **NPN** | nonparanormal |
| **MMS** | minimum model size |
| **PH** | proportional hazards |
| **PO** | proportional odds |
| **NOSF** | number of selected features |
| **METABRIC** | Molecular Taxonomy of Breast Cancer International Consortium |
| **NCDB** | National Cancer Database |
| **GEO** | Gene Expression Omnibus |

## ACKNOWLEDGEMENTS

We are very grateful to the associate editor and the referees for their very valuable comments and suggestions that helped to improve the manuscript.

### Funding

This work was supported by the grant MOST 110-2118-M-035-001-MY2 from the Ministry of Science and Technology of Republic of China (Taiwan). The funders had no role in study design, data collection and analysis, decision to publish, or preparation of the manuscript.

### Grant Disclosures

The following grant information was disclosed by the authors:
Ministry of Science and Technology of Republic of China: MOST 110-2118-M-035-001-MY2.

### Competing Interests

The authors declare that they have no competing interests.

### Author Contributions

- Jie-Huei Wang conceived and designed the experiments, performed the experiments, analyzed the data, prepared figures and/or tables, authored or reviewed drafts of the paper, and approved the final draft.
- Cai-Rong Li performed the experiments, analyzed the data, prepared figures and/or tables, and approved the final draft.
- Po-Lin Hou performed the experiments, analyzed the data, prepared figures and/or tables, and approved the final draft.

### Data Availability

R codes for the simulation studies and real data are available at Figshare: Wang, Jie-Huei (2021): The R code for the paper entitled "Feature Screening for Survival Trait with

Application to TCGA High-dimensional Genomic Data". figshare. Software. https://doi.org/10.6084/m9.figshare.16677070.v3.

The TCGA ESCA, PAAD, LUAD, and BRCA genomic data with survival traits analyzed during this study are all available at Figshare: Wang, Jie-Huei (2021): The TCGA cancer data for the paper entitled "Feature Screening for Survival Trait with Application to TCGA High-dimensional Genomic Data". figshare. Dataset. https://doi.org/10.6084/m9.figshare.16677619.v2.

## Supplemental Information

Supplemental information for this article can be found online at http://dx.doi.org/10.7717/peerj.13098#supplemental-information.

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
