# Peer review of "Feature screening for survival trait with application to TCGA high-dimensional genomic data"

_PeerJ, doi:10.7717/peerj.13098_

## Round 0.1 · original submission · Major Revisions

Reviewer #2 raised some serious concerns about the evaluation and the evidence you presented. I concur with all of their comments, in particular with the emphasis on the c-index metric and comparison to ground truth data. To be clear, it is not only the design of the evaluation, the technical aspects, and exposition which need to be addressed, but also the general scientific story. Reviewer #2 had questions about the difference from your 2021 publication. I also concur that additional benchmarking focused on your method alone seems derivative. A thorough evaluation of methods including yours, indicating the impact different methods have on the features selected (cf. biology plausibility analysis of features) would be a better study design in my opinion. See also the comments by reviewer #1.

·

Basic reporting

Please see my additional comments.

Experimental design

Please see my additional comments.

Validity of the findings

Please see my additional comments.

Additional comments

The manuscript entitled “Feature Screening for Survival Trait with Application to TCGA High-dimensional Genomic Data” by Jie-Huei Wang et al. systematically reviewed multiple screening approaches and developed an optimal model for patient using TCGA genomic datasets with survival information. The study is of clinical significance with some minor issues.
1. Does each biomarker gene have equal or different weight?
2. How is the performance of biomarker genes identified in the manuscript to predict cancer patients’ prognosis compared to published gene signatures that predict survival?
3. Authors could validate their biomarker genes in other independent human cohorts, like METABRIC.

·

Excellent Review

This review has been rated excellent by staff (in the top 15% of reviews)
EDITOR COMMENT
A fantastic review: thoughtful, detail-oriented, very constructive, particularly when identifying areas which need improvement. Thanks to the reviewer for their great work.

Basic reporting

The authors consider the problem of feature selection in the context of survival analysis using regression models. The authors evaluate network-adjusted feature selection methods against several existing univariate feature selection methods on both simulated and real data.

Overall, I found the paper difficult to follow; reading the paper was much like being a detective trying to piece the evidence together, ensuring that I'm interpreting things as intended, and reverse engineering the story. The paper would benefit from language edits as well as additional explanations.

The paper would benefit from substantial re-organization. As it stands now, the paper is difficult to follow. I would propose moving lines 147-185 to the Methods section. It was not clear to me that the ESCA and PAAD results sections were subsets of the Real Data Application Section. You may consider adjusting the font sizes or indicate this in one way or another. Further, you should provide some introductory text in the section summarizing what data you will be analyzing before discussing the evaluation metrics.

Figures: I am not able to read the figure axis labels due to the low resolution and font sizes. Please increase the resolution and font sizes. Please spell out abbreviations such as MMS , provide a short explanation (e.g., 1 sentence), and indicate how to interpret the metric (e.g., small is better). Please label the subfigures (e.g., a, b, c, etc.) In the set similarity heat maps, the color bars indicate a range of possible values from -1 to 1, but Jaccard similarity only returns values in the range of 0 to 1. Please update the range of the color bars.

The authors should cite sources for metrics such as the c-index.

Experimental design

Overall, I think this work has the potential to be very informative. The overall setup of the experiments makes sense. Overall, the authors’ main challenge is an over-emphasis on comparative (relative) metrics without a clear evaluation against ground truths, especially of the predicted survival times. When applied to the simulated data, the NPN-MB method clearly recovers a larger proportion of the true predictors, results in models with fewer variables, and selects a different subset of features than the other methods. The authors should indicate the impact of the feature selection on the survival time predictions errors. For example, the authors should create additional plots like the first row of Figure 1 giving the c-index as a function of the number of selected features for each method. The authors should also include the ground truth set of predictors in the set similarity analysis of features selected by each method.

That said, it was not clear how this paper is different from the authors' previous paper (Wang & Chen 2021). If I am understanding correctly, the method being evaluated was actually introduced and already published in (Wang & Chen 2021) and evaluated on a subset of the TCGA data. The current manuscript appears to build on the initial work by evaluating the method on additional data from the TCGA data set. More evaluation is good, but the authors should clarify the relationship between the two manuscripts and the intent of the work in the current manuscript.

Validity of the findings

In the applications to real data, the authors evaluate the models based on five metrics, three of which are different ways of testing that the predicted survival times of the groups are different statistically. Those three analyses are somewhat meaningless, however, and thus, distracting. The important metric is the c-index, which compares the ranking of the subjects by their predicted survival times with rankings from their true survival times. For the ESCA TGSA data set, the c-index indicates that the ordinary MCP model (0.8466) using all of the variables is more accurate than any model constructed using the feature selection techniques. Secondly, The c-index score of the NPN-MB method (0.7380) is very similar to that of the CINDEX univariate feature selection method (0.7324). It is not clear that feature selection NPN-MB produces a model with more accurate predictions than other feature selection methods. That said, however, the authors’ main focus does not appear to be the creation of more accurate models but on the selection of informative features. The authors analyze genes identified by their feature selection method and cite independent studies that corroborate some of the results. This analysis indicates that NBN-MB is finding biologically-plausible (meaning, informative) associations. Without a similar analysis of features selected by the other methods, however, we cannot concluder whether NBN-MB is better or worse at finding informative features than any other method.

As it currently stands, I do not believe that the conclusions are well supported. The authors will need to perform a major iteration on their analyses, especially with respect to the choice of interpretation of the chosen metrics. That said, I do think the method is interesting and a valuable contribution, so the evaluation on additional data sets as reported in this paper has potential. With appropriate revisions, I would not hesitate to recommend acceptance.

Additional comments

Thank you for the opportunity to read about your work. It is very interesting! I look forward to reading a revised and improved version of the manuscript.

---

## Round 0.2 · Major Revisions

I would like to commend you on the progress on the manuscript, which has addressed all of reviewer #1 concerns. I do however agree with reviewer #2 that there are further issues that should be addressed pre-publication. Please do address all the remaining concerns indicated by the review and in particular re-organize and clarify the section on results of TCGA data.

"Real Data Application with TCGA PAAD data" and "Real Data Application with TCGA ESCA data" are nearly identical.

I think after addressing the reviewer's concern, outlining the method you follow once, and filling out the different numbers and results for both data sets obtained with the same method would make those sections much easier to read.

·

Basic reporting

Authors have successfully responded to all my comments.

Experimental design

None.

Validity of the findings

None.

Additional comments

None.

·

Basic reporting

The overall structure and organization of the paper has been improved considerably. The paper is much easier to read and clearer. The authors should be commended for their work.

At this point, I am mostly finding small typos and grammatical errors. I would encourage the authors to have someone proofread the manuscript.

The paper does include a large number of figures. Some of these figures can be combined. For example, Figures 1-5 should be combined into a single multi-panel figure and Figures 6-10 should be combined into a second multi-panel figure.

Figures 1 and 2 and then Figures 6 and 7 are somewhat redundant. The overlap coefficient is a a normalized version of the metric from Figures 1 and 2. That said, I think that the overlap coefficient is more informative. You could remove Figures 1 and 6 and specify the total number of true predictors in the figure legends so that the reader has infer the number of selected features from the given overlap coefficients.

Experimental design

The methods and results sections have been vastly improved. It is much easier to read and provides more details necessary to understand and replicate the experiments. The authors should be commended for their improvements there.

I am still struggling with the presentation of the results for the real data. I believe that the evaluations of the simulated and real data should be as consistent as possible.

My understanding is that you are trying to find parsimonious models. Meaning, if you models trained on two sets of features that have the same prediction accuracies, you want to select the model that uses feature features because the smaller number of selected features is easier to interpret / evaluate in follow up studies about biological function.

The choice of metrics and how they are currently presented for the real data does not make it easy to compare the feature selection methods.

My understanding is that the methods are run with a given parameter value to select features. A Cox regression model is trained on the chosen features and the accuracies of the models' predictions are evaluated. Thus, there should be the following possible questions and evaluations:

- How accurate are the survival rate predictions compared with the ground truth values in absolutely terms? E.g., if 80% of the patients survive into month 10, then the model predictions should match that.

- I do not know enough about the data sets being used to know if this can be evaluated.

- How accurate are the survival rate predictions compared with the ground truth values in relative terms? E.g., if patient A survives longer than patient B, then patient A should be ranked above patient B according to the predicted survival time.

- It is my understanding that the c-index metric is the only metric that answers this question. The other metrics (e.g., statistical tests) used only compare how well the models separate the two groups (good, poor) – this is not really useful and can be answered from the c-index metrics if you report the c-index for each class (good, poor) individually and together (so 3 c-index values per feature selection method per data set). The other metrics reported just serve to confuse the reader.

- Likewise, Figures 11 and 12 is not a very useful figure since the ground truth survival curves are not plotted alongside the predicted curves.


- How sensitive and precise is the feature selection?

- E.g., how many of the true predictors are chosen? To adequately evaluate this, you would need to know the true predictors. This is probably not available but if you find papers that support the roles of specific genes, you can at least evaluate how many of those genes are found among the selected features.

- E.g., I can do no feature selection and get perfect sensitivity as above. Ideally, a feature selection method finds all ground truth predictors and none of the other predictors. Again, since you may not have a list of all biologically-relevant predictors, you may only be able to report the number of features selected by each method.

- Both of these metrics (precision, number of selected features) should be reported in tables 2 and 4 alongside the c-index values.

Validity of the findings

The discussion and conclusion section text has also been vastly improved. The authors should be commended for their improvements there.

I believe that the findings (as currently presented) do support the research question. Without the changes I have requested, however, the results for the real data are somewhat confusing and hard to navigate, making it difficult to be sure.

Additional comments

The authors have greatly improved the overall readability of the paper. I enjoyed reading this revised version quite a bit. I think there are still some important changes to make but these are not insurmountable.

---

## Round 0.3 · Minor Revisions

Congratulations, the reviewer is happy with all the changes and all the remaining comments are purely figure quality related. I agree with the request to increase font sizes in labels and the resolution of figures. Please additionally review the instructions for authors wrt figure quality.

·

Basic reporting

- The text (e.g., axis labels) in the figures need to be increased to be legible.
- The DPI of the figure images needs to be increased to make them legible

Experimental design

- no comments

Validity of the findings

- no comments

Additional comments

I want to thank the authors for addressing all of the comments in my previous review. The figures are much easier to interpret and read. The results are much clearer.

---

## Round 0.4 · accepted · Accept

Note that the last revision still does not address the issues with the figures raised by the reviewer and confirmed by myself. Font sizes of labels for example are still too small.

After discussion with the PeerJ editorial office we decided that compliance on these remaining issues will be checked by the production office staff of the journal.